# Synthetic Curcumin Analogues Present Antiflavivirus Activity In Vitro with Potential Multiflavivirus Activity from a Thiazolylhydrazone Moiety

**Mateus Sá Magalhães Serafim** [1], **Thales Kronenberger** [2,3,4], **Renata Barbosa de Oliveira** [5], **Erna Geessien Kroon** [1], **Jônatas Santos Abrahão** [1], **Bruno Eduardo Fernandes Mota** [6,*] **and Vinícius Gonçalves Maltarollo** [5,*]

1    Departament of Microbiology, Instituto de Ciências Biológicas, Universidade Federal de Minas Gerais (UFMG), Belo Horizonte 31270-901, Brazil
2    Institute of Pharmacy, Pharmaceutical and Medicinal Chemistry and Tübingen Center for Academic Drug Discovery, Eberhard Karls University Tübingen, Auf der Morgenstelle 8, 72076 Tübingen, Germany
3    Tübingen Center for Academic Drug Discovery & Development (TüCAD2), 72076 Tübingen, Germany
4    School of Pharmacy, Faculty of Health Sciences, University of Eastern Finland, 70211 Kuopio, Finland
5    Departament of Pharmaceutical Products, Faculdade de Farmácia, Universidade Federal de Minas Gerais (UFMG), Belo Horizonte 31270-901, Brazil
6    Departament of Clinical and Toxicological Analysis, Faculdade de Farmácia, Universidade Federal de Minas Gerais (UFMG), Belo Horizonte 31270-901, Brazil
*    Correspondence: brunofmota@gmail.com (B.E.F.M.); viniciusmaltarollo@gmail.com (V.G.M.)

**Abstract:** Arboviral diseases caused by flaviviruses, such as dengue, are a continuing threat and major concern worldwide, with over three billion people estimated to be living with the risk of dengue virus (DENV) infections. There are thus far no antiviral drugs available for treatment, and limited or no vaccines are available. Curcumin and seven synthetic analogues were evaluated for their antiviral activity against dengue virus serotype 2, yellow fever virus and Zika virus, as well as for their cytotoxicity in Vero cells, both by employing MTT assays. Compounds **6** and **7**, which present a thiazolylhydrazone moiety, showed moderate activity against all three flaviviruses, with selectivity index (SI) values up to 4.45. In addition, the envelope protein (E) was predicted as the potential target inhibited by both compounds, supported by molecular docking and dynamics simulation analysis. We hope that this data can contribute to the development of new curcumin antiviral analogues in the near future and can help in the search for new promising compounds as potential therapeutic agents to treat flaviviruses infections.

**Keywords:** curcumin analogues; dengue virus serotype 2; yellow fever virus; Zika virus



## 1. Introduction

Curcumin is a natural compound extracted from *Curcuma longa* L. with several well-known biological activities such as anti-inflammatory, antioxidant, antimicrobial, and others [1]. For instance, there are reports in the literature of its antiviral effect against RNA arboviruses, such as dengue virus serotype 2 (DENV-2) [2,3], as well as Zika virus (ZIKV) and chikungunya virus (CHIKV) [4].

Among other viral diseases, dengue and Zika were classified as epidemics by the World Health Organization (WHO), with Zika remaining a priority disease for research and development in emergency contexts [5], and dengue as a priority disease for surveillance [6], with over three billion people estimated to be at risk of infection. In addition, to date, there are no specific drugs currently available for the treatment of these diseases. Taking curcumin into consideration in this scenario, this compound's biological potential against different viruses could be considered and further explored towards its therapeutic potential,

as reported from its in vitro activity of a curcumin nanoemulsion against all four DENV serotypes [7].

On the other hand, regardless of its activities, curcumin has been demonstrated to have poor bioavailability due to low aqueous solubility, low absorption, high metabolism rate and rapid clearance [8,9]. It has also been considered as a pan-assay interference compound (PAINS), that is, a compound highly related to false positive results, and even as an invalid metabolic panacea (IMP), that is, a compound from natural sources with high abundance which is over-studied [10]. Altogether, these data must be taken into consideration regarding the use of curcumin itself as a potential therapeutic agent. In this regard, some successful strategies can improve the oral bioavailability of curcumin, such as the co-administration with different agents (e.g., piperine [11]) [12], the use of diversified formulations to increase curcumin's solubility [13] and stability [14], as well as structurally modified compounds [15] from the parent structure [16].

In this context, the curcumin analogues **1–7** (Figure 1), previously synthesized by our research group, were submitted to in vitro assays to evaluate their potential antiviral activity against different flaviviruses. These analogues have already been described as having antiparasitic [17] and anti-inflammatory [18] activities.

**Figure 1.** Chemical structures of curcumin analogues synthesized by our group and reported in the literature: compounds **1–5** [17], and **6** and **7** [19]. Moieties similar to that of curcumin are highlighted in blue, while a thiazolylhydrazone moiety is featured in red.

Compounds **6** and **7** are hybrid analogues with curcumin and thiazolylhydrazone moieties (Figure 1). Thiazolylhydrazone derivatives usually exhibit a broad spectrum of biological activities, such as antifungal [19], antibacterial [20], and anticancer [21]. However, despite their therapeutic potential, the antiviral activity of this class of compounds remains little explored, having been described, for example, as influenza neuraminidase inhibitors [22].

Due to the limited data regarding the use of thiazolylhydrazone derivatives as antiviral compounds, it has become of interest to investigate their potential activity against some flaviviruses, such as DENV, ZIKV and yellow fever virus (YFV), from a hybrid molecule with curcumin. In addition, the employment of computer-aided drug discovery (CADD) techniques, which has been successful in the discovery of new potential antiviral agents in the past (e.g., boceprevir [23] and ritonavir [24]), was employed in this work for the prediction

of similar inhibitors and possible molecular targets regarding the evaluated compounds, aiming to further explore them as potential therapeutic agents in the future [25].

Herein, we present and discuss the antiviral activity of seven curcumin analogues against different flaviviruses, highlighting the chemical structural importance of a thiazolyl-hydrazone moiety. Compounds **6** and **7** were successfully determined with a multiflavivirus potential in MTT assays, and the envelope protein (E) was predicted as a potential target, supported by molecular docking and dynamics simulations.

## 2. Materials and Methods

### 2.1. Synthesis of the Compounds **1**–**7**

Compounds **1**–**7** were synthesized according to a previously published procedure and characterized by their melting point and spectral data (FTIR, $^1$H-NMR and $^{13}$C-NMR) [17,19]. Compounds **1**–**4** were synthesized from an aldol condensation reaction using 2 mmol of corresponding aldehydes (quinoline-4-carboxaldehyde, pyrrole-2-carboxaldehyde, isoquinoline-5-carboxaldehyde or 4-(dimethylamino)benzaldehyde) and 1 mmol of cyclopentanone or cyclohexanone dissolved in 2 mL of EtOH 95%. Then, 1.5 mL of a solution of NaOH 0.2 mol/L was added dropwise and the reaction stirred at room temperature until completion. Compound **5** was synthesized under the same conditions, only changing the proportion between the 4-(dimethylamino)benzaldehyde (1 mmol) and cyclopentanone (2 mmol) [17]. Compounds **6** and **7** were synthesized in two steps. In the first step, 0.150 g (0.78 mmol) of (*E*)-4-(4-hydroxy-3-methoxyphenyl)but-3-en-2-one was reacted with 0.071 g (0.78 mmol) of thiosemicarbazide in the presence of three drops of acetic acid in 5 mL of EtOH 95%. The reaction mixture was kept under magnetic stirring and reflux for 12 h. Then, the compounds **6** and **7** were synthesized by reaction between 0.4 mmol of the thiosemicarbazone obtained in the previous step and 0.4 mmol of 2-bromo-4′-methoxyacetophenone or 2-bromo-4′-chloroacetophenone, respectively, in 10 mL of isopropyl alcohol. The reaction mixture was kept under reflux and magnetic stirring for approximately 2 h [19].

### 2.2. Cell Lines

Vero cells (ATCC® CCL-81™) were maintained in Minimal Essential Medium (MEM, Cultilab, Campinas, Brazil). The medium was supplemented with 5% fetal bovine serum (FBS, Cultilab, Campinas, Brazil), 100 IU/mL of penicillin (Thermo Fischer Scientific, Waltham, MA, USA), 100 μg/mL of streptomycin (Thermo Fischer Scientific, Waltham, MA, USA) and 0.25 μg/mL of amphotericin B (Merck, Darmstadt, Germany).

### 2.3. Viruses

The viruses were kindly provided by the Laboratório de Vírus at Universidade Federal de Minas Gerais (UFMG). Zika virus (PE243) was isolated in Recife, Brazil, from a male patient in 2015, at the Fundação Oswaldo Cruz (FIOCRUZ), as described previously [26]. Dengue virus 2 (PI59), lineage II, was identified during an outbreak in Piauí, Brazil, as described previously [27]. The vaccine of yellow fever virus (17DD) was kindly provided by Bio-Manguinhos (FIOCRUZ N. 980FB066Z). All viruses were replicated and titrated in Vero cells. This work is registered in SISGEN, under number A4DCF6B.

### 2.4. Viral Replication

Culture cell flasks of 75 cm$^2$ containing Vero cells were cultivated with MEM with 5% FBS and incubated at 37 °C and 5% CO$_2$ atmosphere. After 24 h, with approximately 60 to 80% confluent monolayer the medium was removed, and the cells were washed three times with phosphate-buffered saline (PBS) to remove cellular debris and serum residue. Cells were infected at a multiplicity of infection (MOI) of 0.01 for each virus (ZIKV, YFV or DENV-2). Adsorption was made for 1 h at 37 °C and 5% CO$_2$ atmosphere with 2 mL of diluted viruses in MEM (without FBS) gently homogenizing the flasks every 10 min. After the adsorption period, 15 mL of MEM with 2% FBS were added to the flasks and incubated under the same conditions. Cell monolayers were observed daily under optical

microscopy until the cytopathic effect was around 80%. The supernatant was then removed and centrifuged at 2016 g in a RT6000B centrifuge (Sorvall, Thermo Scientific, Waltham, MA, USA), for 10 min at 4 °C. Viruses were further stored at −70 °C.

### 2.5. Viral Titration

Virus titration was done in 12-well plates ($1.5 \times 10^5$ cells per well) incubated at 37 °C and 5% $CO_2$ atmosphere for 24 h. Each well received ZIKV, YFV or DENV-2 serial dilutions in MEM (without FBS), to a 1:10 ratio ($10^{-1}$ to $10^{-5}$), following adsorption for 1 h under the same previous conditions. The cells were overlayed with 1.5 mL of M199 medium (Cultilab, Brazil), with 2% FBS and 1% carboxymethylcellulose (CMC) (Synth, Brazil). Plates were incubated for five (ZIKV and YFV) or seven (DENV-2) days at 37 °C and 5% $CO_2$ atmosphere, and then fixed with 10% formalin overnight, gently washed with water and stained with a 1% crystal violet solution for 20 min [28].

### 2.6. Cytotoxicity Assay

Cytotoxicity in mammalian cells was assessed using the MTT tetrazolium reduction assay [29]. Vero cells were seeded in 96-well plates ($4.0 \times 10^4$ cells per well) and incubated at 37 °C and 5% $CO_2$ atmosphere. After 24 h of incubation, 200 μL of fresh MEM medium (1% FBS) containing a serial dilution of the compounds (100 to 1.56 μM) were added to the plates. A serial dilution of DMSO was used as vehicle/viability control, as well as an inhibition control (10% DMSO solution) and a blank (medium only). After 72 h of incubation at 37 °C and 5% $CO_2$ atmosphere, the medium was removed and 100 μL of MTT (3-(4,5-dimethylthiazol-2-yl)-2,5-diphenyltetrazolium bromide) (ThermoFischer Scientific, USA) solution in MEM (0.5 mg/mL) were added to each well. After 3 h of incubation at 37 °C and 5% $CO_2$ atmosphere, the medium was removed and 100 μL of DMSO was added to each well to solubilize formazan crystals. After shaking for 15 min, absorbance at 570 nm of each well was read using a spectrophotometer (VersaMax, Molecular Devices, San Jose, CA, USA). The percentages of inhibition of cell viability were calculated as the ratio between the absorbance of the compound-treated cells and the cells with vehicle only. The 50% cytotoxic concentration ($CC_{50}$) is defined as the lowest concentration of a specific compound that reduces the viability of cultured cells by 50%. Linear regression (LR) was used for the analysis, considering results with $r^2 > 0.9$. All conditions were tested in triplicate and at least two independent assays ($n \geq 6$).

### 2.7. Antiviral Activity Assay

Vero cells were seeded in 96-well plates ($4.0 \times 10^4$ cells per well) and incubated at 37 °C and 5% $CO_2$ atmosphere. After 24 h of incubation, the medium was removed and 100 μL of viral suspensions of ZIKV, DENV-2 and YFV prepared in MEM (1% FBS) to a multiplicity of infection (MOI) of 0.1 were added to the plates. That is, 100 μL of fresh medium containing $4.0 \times 10^3$ viral particles were added to each well. Subsequently, fresh medium containing a serial dilution of compounds (diluted at least eightfold from the $CC_{50}$ values) were added to the plates. Ribavirin was used as an inhibition control. A serial dilution of DMSO was used as a vehicle control. Wells that contained only viruses were used as infection control, and wells without cells as sterility and blank controls. Treatment with MTT follows the same protocol as for $CC_{50}$. The 50% Effective Concentration ($EC_{50}$) was calculated as the percentages of the ratio between the measured absorbance of the compound-treated infected cells and the infected cells with vehicle only. The $EC_{50}$ corresponds to the concentration that inhibits 50% of virus-induced cytopathic effects. The selectivity indexes (SI) were calculated using the ratio between $CC_{50}$ and $EC_{50}$ values. LR was also used for the analysis, considering results with $r^2 > 0.9$. All conditions were tested in triplicate and at least two independent assays ($n \geq 6$).

### 2.8. Target Prediction

*In silico* predictions of molecular targets for active compounds were carried out using the same protocol proposed by Vallone et al., 2018 [30], and Serafim et al., 2019 [31]. Known active compounds against DENV-2, YFV and ZIKV were retrieved from ChEMBL 26 [32]. Compounds with $EC_{50}$ values higher than 50 μM, samples without experimental activity, inactive compounds, and mixtures of compounds were removed from dataset. The lowest energy conformer of all compounds [33] (known actives and compounds identified in this work) were generated using OMEGA 3.1.2.2 software [34], and the active compounds identified in this work were employed as templates for three-dimensional similarity calculations [35] using vROCS 3.3.0.3 software [36]. At this step, a combination of shape and functional group similarities, expressed as TanimotoCombo (TC, ranging from 0 to 2, meaning completely different and similar compounds, respectively), was used to select the most similar compounds from the literature. The known molecular targets of those similar molecules (TC > 1), in other words, higher than 50% similarity, were then suggested to the investigation of putative binding modes of the compounds studied in this work.

### 2.9. Binding Analysis of Ligands by Molecular Docking

The molecular docking studies were carried out using available structures of proposed molecular targets retrieved from Protein Data Bank (PDB) [37], followed by adding hydrogen atoms, removing water molecules, and removing co-crystallized ligands from the structure, and excluding alternate occupancies if needed [38]. Molecular docking was performed around the co-crystallized ligand of the structure deposited under PBD ID 1OKE [39] for DENV using OpenEye docking FRED software [40]. Selection parameters were the following [38]: (i) resolution < 2.0 Å; (ii) presence of co-crystallized inhibitors; (iii) absence of mutations; and (iv) complete chains, disregarding structural modelling. In addition, ZIKV (PDIB ID 5JHM [41]) and YFV (PDB ID 6IW2 [42]) structures were selected for comparison and prepared using Protein Preparation module from Maestro software (Schrödinger Release 2022-3, Schrödinger, LLC, New York, NY, USA) aiming to model missing loops. Furthermore, co-crystallized ligand of the structure PBD ID 1OKE was inserted in both ZIKV and YFV 3D structures followed by energy minimization to relax binding site residues, allowing docking studies. The co-crystallographic ligand and compounds **6** and **7** structures were prepared using OMEGA 3.1.2.2 software with the same previously described protocol, keeping at most 30 of the lowest energy conformers for each ligand. Molecular docking simulations were carried out using a binding site defined as a cubic box of approximately 1300 Å$^3$ around the co-crystallized ligands, and the poses were ranked according to FRED Chemgauss4 score. Visual inspection was also carried out to select most representative poses.

### 2.10. Molecular Dynamics Simulations

Homodimeric MD simulations were performed employing Desmond [43] with the OPLS4 force-field [44,45] for each proposed ligand and for co-crystallized ligands, as well as for apostructures of each target. The simulations encompassed protein–ligand complexes, a predefined water model (TIP3P [46]) as solvent, and counterions (Na$^+$ or Cl$^-$ adjusted to neutralize the overall system charge). The system was treated in an orthrombic box with periodic boundary conditions specifying the shape and the size of the box (13 × 13 × 13 Å) as distance from the box edges to any atom of the protein. A time step of 1 fs was employed to all simulations, with short-range coulombic interactions treated (cut-off value of 9.0 Å) using the short-range method, whilst Smooth Particle Mesh Ewald method (PME) for long-range coulombic interactions [47].

Initially, relaxation of the system was performed with Steepest Descent and the limited-memory Broyden–Fletcher–Goldfarb–Shanno algorithms (Desmond standard settings). The simulation was performed under the NPT ensemble for 5 ns implementing the Berendsen thermostat and barostat methods [48] during the equilibration step. A constant temper-

ature of 310 K was kept throughout the simulation using the Nose–Hoover thermostat algorithm [49] in addition to the Martyna–Tobias–Klein Barostat [50,51] algorithm to maintain 1 atm of pressure. A single production step of at least 100 ns was performed after the systems' minimization and relaxation, with frames being recorded/saved every 1000 ps. Five independent replicas were run, totalling around 500 ns per system simulation.

Representative structures were selected by root-mean-square deviation (RMSD) changes, that is, a representative frame was selected at random points of the trajectory where RMSD were not fluctuating (after the systems' equilibration). In addition, changes in the root-mean-square fluctuation (RMSF) were normalized by residue for the protein backbone. Furthermore, the molecular mechanics-generalized Born surface area (MM-GBSA) method [52,53] was employed to calculate binding energies for ligand–protein complexes. Energy measurements, trajectories, and interaction data are available are available in separated reports in the Zenodo repository (under the codes: 10.5281/zenodo.663731). Lastly, protein–ligand interactions were determined using the Simulation Event Analysis pipeline implemented in Maestro (Schrödinger Release 2022-4, Schrödinger, LLC, New York, NY, USA).

## 3. Results

### 3.1. Antiviral Activity of Curcumin Analogues

The cytotoxic concentration of 50% ($CC_{50}$) in Vero cells was determined by employing MTT in 96-well microplates. The compounds showed $CC_{50}$ values ranging from <6.25 to >100 μM (Table 1). The effective concentration of 50% ($EC_{50}$) was also assessed with MTT under the same conditions, adding a viral suspension (MOI 0.1) of ZIKV, YFV or DENV-2. Curcumin and three compounds, **2**, **6** and **7,** presented antiviral activity for at least one of the three flaviviruses, with **6** and **7** being active against all three viruses. As for ZIKV, $EC_{50}$ values ranged from 4.04 ± 0.38 to 32.45 ± 1.58 μM, resulting in selectivity indexes (SI) from 4.45 to 1.33, respectively. Compounds **6** and **7** were active against YFV, with $EC_{50}$ values of 10.85 ± 0.13 and 11.94 ± 0.39 μM, and SI values of 1.63 and 1.51, respectively. Lastly, curcumin (13.63 ± 0.95 μM), **6** (12.5 μM) and **7** (15 μM) were active against DENV-2, with SIs of 1.67, 1.42 and 1.2, respectively. Results are summarized in Table 1.

**Table 1.** Antiviral activity (μM) against three flaviviruses: ZIKV, YFV and DENV-2.

| Compounds | $CC_{50}$ (μM) | ZIKV (PE243) | | YFV (17DD) | | DENV-2 (PI59) | |
|---|---|---|---|---|---|---|---|
| | | $EC_{50}$ (μM) | SI [a] | $EC_{50}$ (μM) | SI [a] | $EC_{50}$ (μM) | SI [a] |
| Curcumin | 22.76 ± 2.84 | 17.12 ± 1.2 | 1.33 | - | - | 13.63 ± 0.95 | 1.67 |
| **1** | <6.25 | - | - | - | - | - | - |
| **2** | >100 | 32.45 ± 1.58 | >3.08 | - | - | - | - |
| **3** | <6.25 | - | - | - | - | - | - |
| **4** | >100 | - | - | - | - | - | - |
| **5** | >100 | - | - | - | - | - | - |
| **6** | 17.7 ± 0.7 | 8.61 ± 0.41 | 2.06 | 10.85 ± 0.13 | 1.63 | 12.5 * | 1.42 |
| **7** | 17.98 ± 2.47 | 4.04 ± 0.38 | 4.45 | 11.94 ± 0.39 | 1.51 | 15 * | 1.2 |
| Ribavirin | >100 | 4.1 ± 0.35 | >24.39 | 40.9 ± 3.49 | >2.44 | - | - |

[a] Selectivity Index: Ratio between $CC_{50}$ and $EC_{50}$ ($CC_{50}/EC_{50}$). $CC_{50}$ and $EC_{50}$ values are represented as the average and standard deviation of the mean calculated from at least two independent experiments, each performed in triplicate ($n \geq 6$). * Single triplicate assay.

### 3.2. Target Prediction, Molecular Docking and Dynamics Simulations

No similar compound (TC > 1) with experimental activity for yellow fever virus was found by comparing compounds **6** and **7**. In addition, there were no hits from the Zika virus known inhibitor database for compounds **6** and **7**. One known DENV-2 inhibitor was selected as similar to compound **6** and two others were selected as similar to compound **7** (Table 2), all of them also being thiazolylhydrazones derivatives designed as E protein inhibitors [54].

**Table 2.** List of most similar compounds from ChEMBL with known molecular targets and selected PDB entries for docking studies.

| ChEMBL ID | TanimotoCombo | | Related Target | Reference | PDB ID |
|---|---|---|---|---|---|
| | 6 | 7 | | | |
|  CHEMBL3401565 | 1.08 | - | E protein | Jadav et al. 2015 [54] | 1OKE |
|  CHEMBL3401563 | - | 1.03 | | | |
|  CHEMBL3401564 | - | 1.03 | | | |

After target prediction, docking studies indicated that compounds **6** and **7** bind to the E protein from the three studied viruses' binding site similarly to the n-octyl-beta-D-glucoside (BOG, the ligand experimentally co-crystallized) (Figure 2A–H). Furthermore, the overall folding of the E protein from DENV-2, ZIKV, and YFV viruses are very similar (Figure 2I), suggesting that the binding mode of common inhibitors could be similar as well. For the DENV-2 E protein system, the carbohydrate moiety of BOG forms hydrogen bonds with Glu-49, Gln-200, Gln-271, and Ala-280, while the hydrocarbon part of this ligand fits a hydrophobic pocket (Figure 2A), which was also predicted properly by the docking protocol (Figure 2B). Interestingly, thiazolylhydrazones (**6** and **7**) form hydrogen bonds with Gln-200 and Gln-271 as well as an additional hydrogen bond with Ala-50 (Figure 2C,D). In addition, the curcuminoid moiety fits the same hydrophobic pocket of n-octyl from the crystallographic ligand (Figure 2C,D).

Interestingly, compounds **6** and **7** can interact with entrance residues at the binding site of ZIKV and YFV E protein, equivalent to the binding mode at DENV-2 E protein pocket (Figure 2E–H). For instance, thiazolylhydrazones form H-bonds with Tyr-203 and Glu-276 from ZIKV E protein, and H-bonds with Ser-202, Lys-266 and Tyr-274, as well as an additional interaction with Val-38 in the hydrophobic pocket of YFV E protein. These poses display transient interactions in all simulated systems (Figure 3A,B), with the thiazolylhydrazones being stabilized by water-mediated hydrogen bonds, originally not predicted by the docking. Most of the trajectories displayed a bimodal distribution of the ligand's RMSF, which suggests a movement within the pocket and instability (Figure 3C). However, in comparison to the BOG, the compounds stay for at least half of the simulation

time within a < 2.5 Å (Figure 3C), allowing us to utilize these trajectories for further studies. MD trajectories were utilized to infer the compounds' predicted binding energy using the MM/GBSA method, suggesting from the median values of the violin plot that both compounds **6** and **7** have a higher affinity to YFV and ZIKV protein E, compared to DENV. These calculated energies corroborate experimental data in which compounds **6** and **7** showed lower activities against DENV in comparison to the other flaviviruses (Table 1).

Additionally, BOG simulations show an unstable binding mode of the original ligand within E protein's dimer binding pocket (Supporting Information Figure S1), with few conserved interactions despite the low variation in the ligand position, as observed by the low RMSD values from the ligand ~3 Å (Figure 3C). Interestingly, compounds **6** and **7** binding mode reproduced some of these relevant interactions (Thr-48 and Ala-50), while expanding on the pocket occupancy and reaching, for example, Thr-280 (Figure 3B,C).

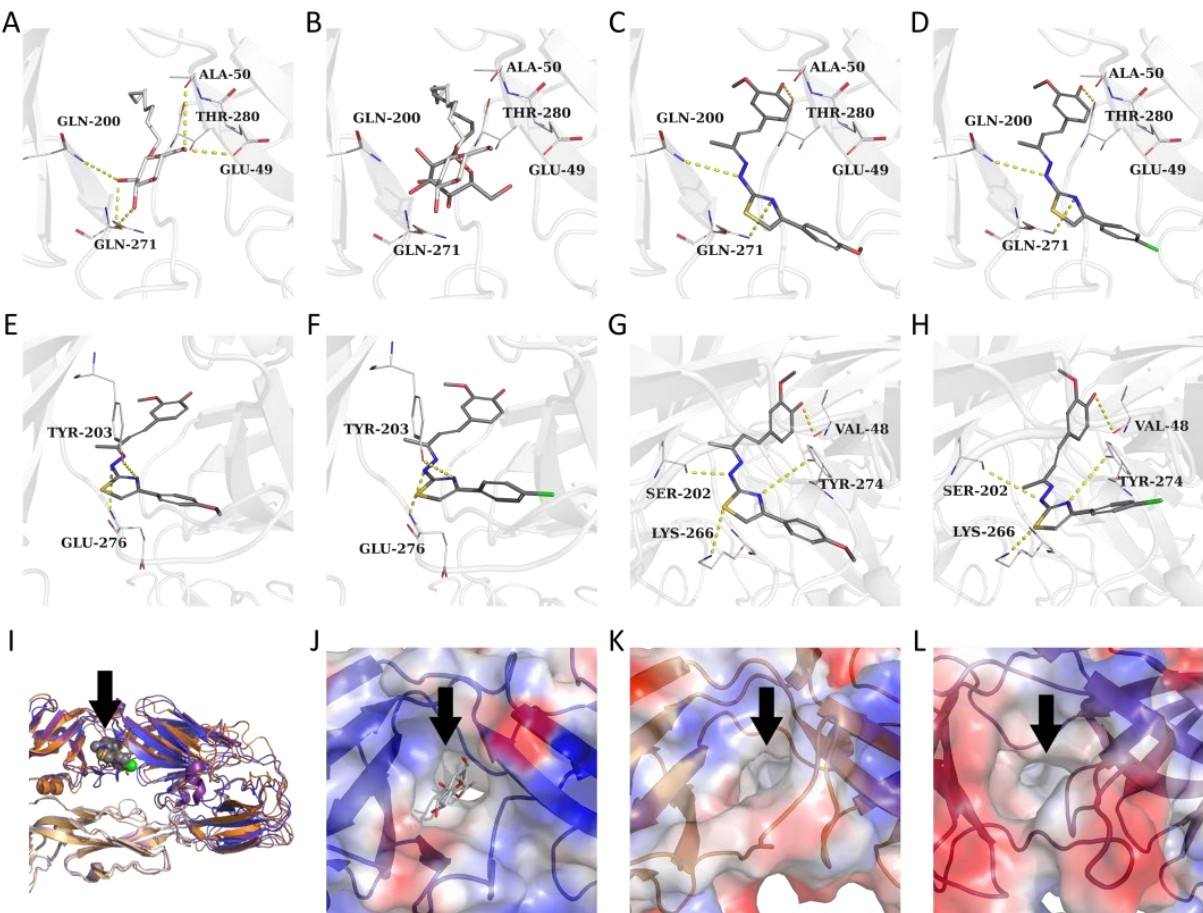

**Figure 2.** Docking studies of the co-crystallized ligand, and compounds **6** and **7**. Experimental binding mode of n-octyl-beta-D-glucoside (BOG) bound to DENV-2 E protein (**A**). Redocking comparison of putative (carbons in dark grey) and experimental BOG binding mode (**B**). Putative binding modes of compounds 6 and 7 in DENV-2 E protein (**C** and **D**, respectively), ZIKV E protein (**E** and **F**, respectively), and YFV E protein (**G** and **H**, respectively). Yellow dashes indicate polar interactions with a range shorter than 4 Å. Superimposition of three viruses' E proteins: DENV-2 E protein in blue, ZIKV E protein in orange, and YFV E protein in purple (**I**). The electrostatic potential (ESP, ranging from red indicating negative regions to blue, positive regions) surface of the binding site E proteins from DENV-2, ZIKV, and YFV (**J**, **K** and **L**, respectively). Black arrows point towards the common binding site of three proteins.

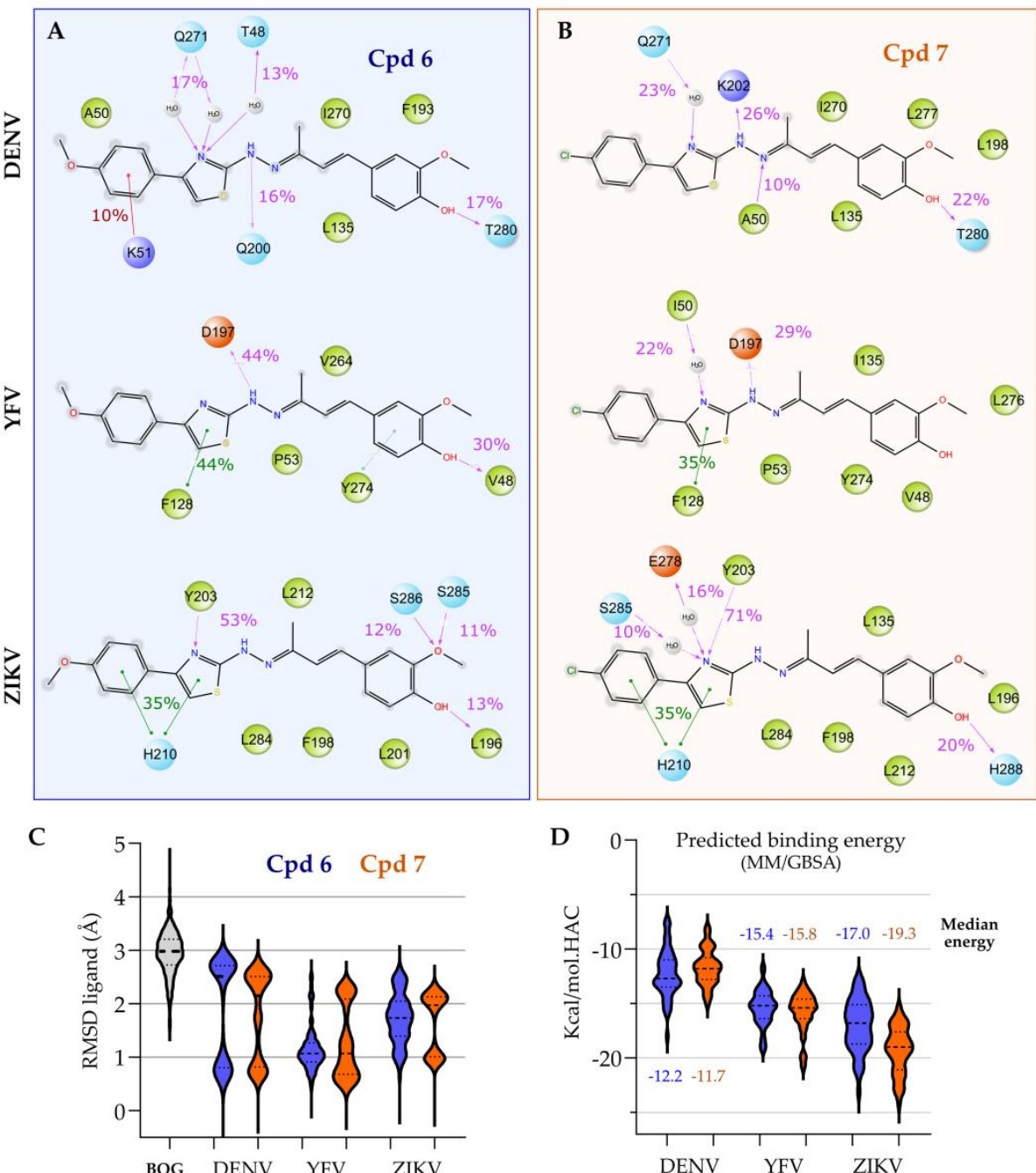

**Figure 3.** Summary of the interactions along the MD simulations. Interaction pattern during the simulation for ligands **6** (**A**) and **7** (**B**), respectively, for each of the simulated systems (5× 100 ns). Pink solid arrows represent interactions with the main chain of the amino acids, while dashed pink arrows represent polar contacts with their side chains. Pi-pi interactions are represented by green arrows. The respective frequency of the observed interaction is depicted as a number above the arrow. Violin plot depicting the variation of ligand's RMSD, calculated from its heavy atoms (**C**). MD trajectories were utilized to infer the compounds predicted binding energy (Kcal/mol.HAC, where HAC—heavy atom count), suggesting from the median values of the violin plot (**D**) that both cpd **6** and **7** have a higher affinity to ZIKV protein E.

## 4. Discussion

Herein, in exception to compound **2**, the low SI values obtained could be troublesome to the applicability of these compounds as potential therapeutic agents in a drug develop-

ment. An ideal drug candidate prototype, for example, should present SI values equal to or greater than 10 [55], such as that observed for ribavirin against ZIKV (>24.39). Ribavirin is a broad-spectrum antiviral drug with an unclear mode of action against some flaviviruses, such as DENV [56,57] and ZIKV [58]. In addition to the proposed inhibition of viral RNA polymerases, impacting host cell and viral gene expression, and acting as a mutagen [58], although potentially concentration dependent [59], ribavirin can also delay the onset of viral replication [57].

Here, lower SI values were obtained for ribavirin against YFV, suggesting those differences in the mode of actions. Low SI values obtained from the proposed compounds in this work have also been shown in similar studies considering curcumin analogues against ZIKV [4], such as bisdemethoxycurcumin and demethoxycurcumin (SI values of 4.43 and 2.23, respectively). Interestingly, however, the two compounds with the thiazolylhydrazone moiety could be thought to focus for higher $CC_{50}$ values, for example, by incrementing their curcumin moiety (e.g., symmetrical structures), such as with compounds **2**, **4** and **5**, which showed $CC_{50}$ values > 100 μM.

The low toxicity of asymmetric compounds observed is somewhat supported by the cytotoxicity of other asymmetrical synthetic curcumin analogues. For instance, 10 asymmetrical mono-carbonyl analogues from curcumin were obtained, and also shown low $CC_{50}$ values (3.94 ± 0.07 to 16.15 ± 0.18 μM) in Vero cells [60], as have some of the compounds in this work. On the other hand, five symmetrical structures proposed by Balasubramanian et al. (2019), were shown to have higher $CC_{50}$ values (25.50 ± 2.64 to 87.40 ± 9.03 μM) in BHK-21 cells (kidney epithelial cells such as Vero), and higher SI values (3.51 to 16.27) when assessed against DENV-2 [61]. Thus, one could argue that a hybrid symmetrical structure with a thiazolylhydrazone substituent could favour both lower cytotoxicity and even a higher activity against DENV-2 and other flaviviruses, similarly to modifications shown by thiazole-coumarin derivatives as potential DENV serine protease inhibitors [62].

It is also noteworthy that large substituents, such as dipeptides, fatty acids, and folic acid in symmetrical curcumin analogues, may be troublesome for cytotoxicity, as shown by five curcumin synthetic bioconjugates employing a similar protocol with MTT and Vero cells, which presented $CC_{50}$ values from > 0.011 to 0.147 μM [63]. Similarly, modifications of substituents over aromatic moieties in curcumin analogues could favour and enhance antiviral activity, as shown in the optimization of E protein inhibitors against DENV-2 [54]. Thiazolylhydrazone substituents have also shown other biological activities in the past, including antibacterial [64], antifungal [65], as well as anti-inflammatory [66] and antioxidant [67]. Additionally, the synthetic curcumin analogues with thiazolylhydrazone moieties proposed in this work could even be supported by the targets predicted from inhibitors' similarities, which were also available in different active compounds with designed thiazolylhydrazone moieties assessed by Jadav et al. 2015 [54] against DENV-2. It is also important to consider that not only can thiazolylhydrazone substituents be considered in novel analogues [62] enhancing biological activity, but also modifications could be added with an aim to reduce cellular toxicity [60], for example, by proposing and obtaining novel hybrid thiazolylhydrazone curcumin compounds. It is important to mention that the employed computational pipeline was based on a state-of-art protocol to target prediction [38,68], comprising molecular similarity followed by molecular docking, and was subsequently evaluated by molecular dynamics simulations. Although those results were consistent, experimental validation by biochemical assays or other experiments is required to determine the binding of compounds **6** and **7** to flavivirus' protein E.

In this sense, considering the potential multiflavivirus activity of compounds **6** and **7**, one could argue that these may be helpful as leads for drug design, considering the structural modifications and synthesis of analogues. Furthermore, specific chemical modifications towards more hydrophilic substituents could potentially enhance curcumin's low aqueous solubility and absorption, as mentioned [8,9], and even enhance the antiviral activity, as shown with the assessment of various hydrophilic substituents (nitrogen nucleophiles) at the same chemical structure position against DENV-2 [69]. These could

then be considered, for example, in the functionalization of food ingredients, for delivering bioactive compounds aiming to mitigate the harmful effects of certain diseases [38], and in reaching out to clinics, as an available therapeutic option to treat flavivirus infections, thus tackling the current scenario.

Lastly, the effects of curcumin treatment against other enveloped viruses, such Japanese encephalitis virus (JEV) and influenza virus were also demonstrated [70]. Authors have indicated that curcumin disrupts the integrity of the membranes of viral envelopes, supporting targets from the compounds similarities analysis (Table 2). This disruption could be associated with the hydrophobic characteristic of lipid membranes, favouring the intercalation of curcumin in the lipid bilayer [70]. One could argue that it could also be associated with cellular toxicity, potentially intercalating in the lipid membrane of host cells. In this sense, compounds designed to increase selectivity to the viral E protein intercalation could be considered.

Once intercalated to the viral target, inhibition could be sustained by hydrogen bonds in between curcumin compounds and specific residues from a protein [71], such as the ones observed with Glu-49, Gln-200, Gln-271, and Ala280 for DENV E protein, as well as Tyr-203 and Glu-276 for ZIKV, and Val-38, Ser-202, Lys-266 and Tyr-274 for YFV. Interestingly, some of these residues were also supported by polar contacts predicted from the molecular dynamics simulations (Figure 3C,D).

Moreover, curcumin has also been shown to have the potential to inhibit the entry of various hepatitis C virus (HCV) genotypes into human hepatocytes by modifying the membrane fluidity of the HCV envelope, inhibiting binding and fusion with the host cell membranes, thus being associated with its E protein [72]. Here, one could argue that this could be somewhat similar to the predicted targets, as HCV is also a member of the *Flaviviridae* family, as are YFV and DENV, and presents some structural similarity and conservation of the E protein [73,74], potentially leading to further exploration of this specific target for these and other curcumin analogues [75] in the future.

## 5. Conclusions

Taken together, these results demonstrate the feasibility of curcumin analogues synthesis in the discovery of novel antiviral agents, including the potential for multiflavivirus activity from a thiazolylhydrazone moiety. In addition, the employment of computational approaches demonstrated viable options to support biological activity, thereby corroborating data available from other compounds in the literature. Here, three out of seven compounds synthetized showed moderate activity against ZIKV, YFV and DENV-2, two of which were active against all three flaviviruses. Molecular docking showed similar binding modes between compounds **6** and **7**, in comparison to the co-crystallized ligands, and their binding mode stability was supported by molecular dynamics simulations. Going forward, we intend to investigate the activity of intermediates in synthesis and other analogues of the selected compounds, aiming to obtain higher SI values and to evaluate them against other DENV serotypes. We hope that this work may also contribute towards narrowing the design of curcumin analogues in the near future, especially considering thiazolylhydrazone substituents, in order to enhance their biological activity and to help in the search for new promising compounds as potential therapeutic agents to treat flavivirus infections.

**Supplementary Materials:** The following supporting information can be downloaded at: https://www.mdpi.com/article/10.3390/futurepharmacol3020022/s1, Figure S1: Summary of the interactions along the MD simulations. (A) Co-crystallized *n*-octyl-beta-D-glucoside (BOG) retrieved from the PDB 1OKE. Pink solid arrows represent interactions with the main chain of the amino acids, while dashed pink arrows represent polar contacts with their side chains. Pi–pi interactions are represented by green arrows. The respective frequency of the observed interaction is depicted as a number above the arrow.

**Author Contributions:** Conceptualization, R.B.d.O., B.E.F.M. and V.G.M.; methodology, M.S.M.S., T.K., E.G.K., J.S.A. and V.G.M.; software, T.K. and V.G.M.; validation, M.S.M.S., T.K. and V.G.M.; formal analysis, M.S.M.S. and V.G.M.; investigation, M.S.M.S., B.E.F.M. and V.G.M.; resources, R.B.d.O., E.G.K., J.S.A. and B.E.F.M.; data curation, M.S.M.S. and B.E.F.M.; writing—original draft preparation, M.S.M.S., R.B.d.O. and V.G.M.; writing—review and editing, M.S.M.S., T.K., J.S.A., B.E.F.M. and V.G.M.; visualization, R.B.d.O., B.E.F.M. and V.G.M.; supervision, B.E.F.M. and V.G.M.; project administration, R.B.d.O., B.E.F.M. and V.G.M.; funding acquisition, R.B.d.O., E.G.K., J.S.A., B.E.F.M. and V.G.M. All authors have read and agreed to the published version of the manuscript.

**Funding:** This research was funded by CAPES foundation, grant numbers 88882.348380/2010-1 and 88887.595578/2020-00; CNPq, grant numbers 428054/2016-1, 132732/2018-1, 302081/2018-6 and 315750/2020-0; FAPEMIG, grant numbers 27103, PPM-00452017, PPM-00417-17 and APQ-03116-18; and UFMG intramural funds. E.G.K., J.S.A. and R.B.O. CNPq researchers. T.K. is funded by the fortune initiative and from T*ü*CAD2.

**Institutional Review Board Statement:** Not applicable.

**Informed Consent Statement:** Not applicable.

**Data Availability Statement:** Energy measurements, trajectories, and interaction data are available are available in separated reports in the Zenodo repository (under the codes: 10.5281/zenodo.663731).

**Acknowledgments:** The authors would like to thank OpenEye Scientific for the academic licenses and UFMG for intramural funds. The authors would like to thank the CSC—Finland for the generous computational resources provided. T*ü*CAD2 is funded by the Federal Ministry of Education and Research (BMBF) and the Baden-Württemberg Ministry of Science as part of the Excellence Strategy of the German Federal and State Governments EXC 2180–390900677. All authors agree with declared acknowledgments.

**Conflicts of Interest:** The authors declare no conflict of interest.

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
