# Peer review of "Synthetic Curcumin Analogues Present Antiflavivirus Activity In Vitro with Potential Multiflavivirus Activity from a Thiazolylhydrazone Moiety"

_futurepharmacol, doi:10.3390/futurepharmacol3020022_

Round 1

Reviewer 1 Report

work is a very interesting research study on the contribution to the development of new antiviral analogs of curcumin that in the near future may help lead to new promising compounds as potential therapeutic agents for the treatment of flavivirus infections.

Author Response

We appreciate the commentaries about our work. Thank you for taking your time reading and reviewing the manuscript.

Reviewer 2 Report

The authors have investigated the role of coumarin derivatives against viral infection like Zika, Dengue, and Yellow fever through in vitro methods. The work done is good and could have possible therapeutic significance. Here some comments on the manuscript.

·         Can the synthesis procedure be described in detail? If possible.

·         Was there any possibility to conduct in vivo animal study for the same compounds against the said viral infections?

·         The mechanism and information related to ribavirin can be introduced and discussed briefly.

·         Can the Conclusion section be more concise?

·         If there are any limitations of the study, it can be mentioned.

·         Future perspective of the compounds can also be given.

·         What could be the clinical and nutritional significance of the compounds?

If appropriate, some of the older references can be updated with recent ones.

Author Response

The authors have investigated the role of coumarin derivatives against viral infection like Zika, Dengue, and Yellow fever through in vitro methods. The work done is good and could have possible therapeutic significance. Here some comments on the manuscript.

Answer: Thank you for taking your time reading and reviewing our work. We appreciate the commentaries and suggestions. All were addressed accordingly to the manuscript, and each one was highlighted throughout the text and down below.

  • Can the synthesis procedure be described in detail? If possible.

Answer: A brief detailed description of the methods was added to the methods section.

  • Was there any possibility to conduct in vivoanimal study for the same compounds against the said viral infections?

Answer: Unfortunately, not at the time. Pandemic laboratory restrictions limited even working with some following in vitro studies. These and following in vivo studies are being considered to an upcoming work, and new structural modifications are in progress. 

  • The mechanism and information related to ribavirin can be introduced and discussed briefly.

Answer: Thank you for the suggestion. A brief discussion about ribavirin mechanism was added to the manuscript.

  • Can the Conclusion section be more concise?

Answer: Thanks for the recommendation. We reworked the conclusion to a more concise version.

  • If there are any limitations of the study, it can be mentioned.

Answer: Thanks for the suggestion. We included a sentence at the discussion section highlighting that simulations does not confirm the mechanism of action, just suggest (based on statistical and physicochemical evidences) a potential path to be further investigated.

  • Future perspective of the compounds can also be given.

Answer: Thanks for the suggestion. A brief perspective of the compounds was added to the manuscript, apart from that of symmetric and asymmetric structures already included.

  • What could be the clinical and nutritional significance of the compounds?

Answer: Thanks for the question. We added a brief discussion regarding clinical and potential nutritional implementation for the compounds.

  • If appropriate, some of the older references can be updated with recent ones.

Answer: Thanks for the recommendation. Some older references were updated and newer ones were added accordingly throughout the manuscript. A few, being the original research to topic discussed, were maintained.

Reviewer 3 Report

The authors present the activity of curcumin and seven analogues synthesized by them, on Vero cells infected by DENV2,ZIKA and YFV. They used ribavirin, a compound acting as a polymerase inhibitor, as a control. In order to determine how useful these compounds are, they obtain a measure of the toxicity over non-infected cells (CC50) and of their activity as antivirals (EC50); the ratio of these is the selectivity index, and it is deemed to be good if it is greater than 10. Based on structural similarity of two of their compounds to other anti flaviviral compounds, the E protein is proposed as a target. Docking is carried out over a pre-determined binding pocket, and the complexes were simulated with molecular dynamics.

The authors failed in communicating the novelty of their compounds; they are derived from the curcumin scaffold, which has been used previously, and the best compounds are a mixture of the curcumin scaffold with a thiazolylhydrazone, a moiety that has also been intensively explored as inhibitor of many different activities.

Regarding the activity of the tested compounds, compounds 6 and 7 are indeed active against the three tested viruses, but their selectivity index is poor. There is no analysis, based on their structures, as to what could be making them toxic to the cells at concentrations that are close to those needed to be antiviral. Given the many known/proposed targets of these moieties, there are many possible reasons for their toxicity.

Regarding the choice of protein E as a target, it is based on reference 47, which is based on crystal structure 1OKE, a dimer of DENV E together with a detergent molecule placed in a hydrophobic pocket. Docking was directed to this pocket, and was carefully validated by redocking the original BOG ligand. Docking compounds 6 and 7 to this same site does not validate protein E as their target. It just shows that it is possible to place the compounds there. Furthermore, the MD simulations have high values of RMSD for the ligands (if the reported RMSD speaks to the variation of position of the ligand with respect to a fixed protein, 3Å is not a small number), suggesting that these poses are not very stable. I tried to access the zenodo data, but it is not accessible. Without that, it is impossible to judge stability or any other feature of the runs.

Given that compounds 6 and 7 are active against the three tested viruses, and that the structure for the Zika and YFV E proteins is also known, I would strongly suggest modeling the interactions of these compounds with Zika and YFV E, analyzing the conservation of the binding site. The rationale for this is that the compounds are active on the three of them, and the proposed mechanism is binding to E. Hence, they should bind equivalently to all three proteins, making the proposal of the mechanism of action more robust. This is particularly important given the alternative explanation for the mechanism of action of this class of poorly water-soluble molecules, as membrane structure disruptors.

Author Response

The authors present the activity of curcumin and seven analogues synthesized by them, on Vero cells infected by DENV2,ZIKA and YFV. They used ribavirin, a compound acting as a polymerase inhibitor, as a control. In order to determine how useful these compounds are, they obtain a measure of the toxicity over non-infected cells (CC50) and of their activity as antivirals (EC50); the ratio of these is the selectivity index, and it is deemed to be good if it is greater than 10. Based on structural similarity of two of their compounds to other anti flaviviral compounds, the E protein is proposed as a target. Docking is carried out over a pre-determined binding pocket, and the complexes were simulated with molecular dynamics.

Answer: The commentaries about our work are very much appreciated. First, we would like to thank the reviewer for taking time reading and reviewing this manuscript. Each suggestion and comment were individually addressed, and modifications were made throughout the text, including the addition of molecular docking and dynamics simulations studies against YFV and ZIKV’s E proteins.

The authors failed in communicating the novelty of their compounds; they are derived from the curcumin scaffold, which has been used previously, and the best compounds are a mixture of the curcumin scaffold with a thiazolylhydrazone, a moiety that has also been intensively explored as inhibitor of many different activities.

Answer: Thanks for the commentary. We have reworked the section regarding the novelty of these compounds in comparison to known or available data discussed.

Regarding the activity of the tested compounds, compounds 6 and 7 are indeed active against the three tested viruses, but their selectivity index is poor. There is no analysis, based on their structures, as to what could be making them toxic to the cells at concentrations that are close to those needed to be antiviral. Given the many known/proposed targets of these moieties, there are many possible reasons for their toxicity.

Answer:  Thanks for the commentary. We added a discussion for the compounds’ toxicity, in order to address their antiviral potential over cytotoxicity.

Regarding the choice of protein E as a target, it is based on reference 47, which is based on crystal structure 1OKE, a dimer of DENV E together with a detergent molecule placed in a hydrophobic pocket. Docking was directed to this pocket, and was carefully validated by redocking the original BOG ligand. Docking compounds 6 and 7 to this same site does not validate protein E as their target. It just shows that it is possible to place the compounds there. Furthermore, the MD simulations have high values of RMSD for the ligands (if the reported RMSD speaks to the variation of position of the ligand with respect to a fixed protein, 3Å is not a small number), suggesting that these poses are not very stable. I tried to access the zenodo data, but it is not accessible. Without that, it is impossible to judge stability or any other feature of the runs.

Answer: We appreciate the commentaries and thank you for the suggestion. We further discussed the dynamics simulations and binding data and added docking studies to both YFV and ZIKV’s E proteins, to complement the available information. Furthermore, we calculated MM-GBSA binding energies to complement the discussion and add an energetic perspective besides the structural interpretation. In addition, we apologize for the issue with the zenodo link. This was also fixed.

Given that compounds 6 and 7 are active against the three tested viruses, and that the structure for the Zika and YFV E proteins is also known, I would strongly suggest modeling the interactions of these compounds with Zika and YFV E, analyzing the conservation of the binding site. The rationale for this is that the compounds are active on the three of them, and the proposed mechanism is binding to E. Hence, they should bind equivalently to all three proteins, making the proposal of the mechanism of action more robust. This is particularly important given the alternative explanation for the mechanism of action of this class of poorly water-soluble molecules, as membrane structure disruptors.

Answer: Thanks for the recommendation. YFV and ZIKV’s E proteins docking and molecular dynamics were added to the manuscript.

Reviewer 4 Report

I am happy to recommend the manuscript titled “Synthetic curcumin analogues present antiflavivirus activity in vitro with potential multiflavivirus activity from a thiazolylhydrazone moiety” by Serafim et al. for publication in Future Pharmacology. The exciting finding reported in the manuscript is supported by computational and scientific experimental data, and the overall content of this manuscript is of high value to Future Pharmacology readers. Curcumin and related compounds are very well-known to the scientific community for their wide range of activity, but none of the compounds are made to clinics. Compounds 6 and 7 displayed comparable activity (EC50) against ZIKV and four-fold improved EC50 against YFP than the reference compound ribavirin. The authors made valid statements that obtaining the optimum selectivity index (SI) could be challenging; in the current study, authors obtained a poor selectivity index for curcumin, compound 6 and 7. Figure 3, Summary of the interactions along the MD simulations; why the geometry of compounds 6 and 7 is changed to Z from E?

Author Response

These commentaries about our work are very much appreciated. We sincerely thank you for taking your time reading and reviewing the manuscript. Compounds’ geometry was fixed in figures as well as in simulations.

Round 2

Reviewer 3 Report

The authors answered all the queries I had. It is a nice surprise to see that the simulations predict differences in affinity that mirror the experimental data.

I suggest that the authors use a native speaker to smooth out some language issues.